# New Species of *Diaporthales* (*Ascomycota*) from Diseased Leaves in Fujian Province, China

**DOI:** 10.3390/jof11010008

**Published:** 2024-12-26

**Authors:** Xiayu Guan, Taichang Mu, Nemat O. Keyhani, Junya Shang, Yuchen Mao, Jiao Yang, Minhai Zheng, Lixia Yang, Huili Pu, Yongsheng Lin, Mengjia Zhu, Huajun Lv, Zhiang Heng, Huiling Liang, Longfei Fan, Xiaoli Ma, Haixia Ma, Zhenxing Qiu, Junzhi Qiu

**Affiliations:** 1Key Laboratory of Ministry of Education for Genetics, Breeding and Multiple Utilization of Crops, College of Horticulture, Fujian Agriculture and Forestry University, Fuzhou 350002, China; gxy302@126.com; 2State Key Laboratory of Ecological Pest Control for Fujian and Taiwan Crops, College of Life Sciences, Fujian Agriculture and Forestry University, Fuzhou 350002, China; mutaichang@163.com (T.M.); 15750811500@163.com (J.S.); maoyuchen301@163.com (Y.M.); jiaoyang0368@163.com (J.Y.); zhengmh0928@163.com (M.Z.); finy1015@163.com (L.Y.); hdpuhuili@163.com (H.P.); linyongsheng0909@163.com (Y.L.); zhumengjia_0529@163.com (M.Z.); 13328626122@163.com (H.L.); 18965917828@163.com (Z.H.); 3Department of Biological Sciences, University of Illinois, Chicago, IL 60607, USA; keyhani@uic.edu; 4Guangxi Institute of Botany, Chinese Academy of Sciences, Guilin 541006, China; llhl@gxib.cn; 5College of Plant Protection, Gansu Agricultural University, Lanzhou 730070, China; fanlf@gsau.edu.cn; 6College of Life Science and Technology, Xinjiang University, Urumqi 830046, China; xlm87@xju.edu.cn; 7Hainan Key Laboratory of Tropical Microbe Resources, Institute of Tropical Bioscience and Biotechnology, Chinese Academy of Tropical Agricultural Sciences, Haikou 571101, China; maihaixia@itbb.org.cn; 8College of Humanities and Law, Fuzhou Technology and Business University, Fuzhou 350715, China

**Keywords:** *Diaporthe*, multigene phylogeny, new species, *Paratubakia*, taxonomy

## Abstract

Fungal biota represents important constituents of phyllosphere microorganisms. It is taxonomically highly diverse and influences plant physiology, metabolism and health. Members of the order *Diaporthales* are distributed worldwide and include devastating plant pathogens as well as endophytes and saprophytes. However, many phyllosphere *Diaporthales* species remain uncharacterized, with studies examining their diversity needed. Here, we report on the identification of several diaporthalean taxa samples collected from diseased leaves of *Cinnamomum camphora* (*Lauraceae*), *Castanopsis fordii* (*Fagaceae*) and *Schima superba* (*Theaceae*) in Fujian province, China. Based on morphological features coupled to multigene phylogenetic analyses of the internal transcribed spacer (ITS) region, the large subunit of nuclear ribosomal RNA (LSU), the partial beta-tubulin (*tub2*), histone H3 (*his3*), DNA-directed RNA polymerase II subunit (*rpb2*), translation elongation factor 1-α (*tef1*) and calmodulin (*cal*) genes, three new species of *Diaporthales* are introduced, namely, *Diaporthe wuyishanensis*, *Gnomoniopsis wuyishanensis* and *Paratubakia schimae*. This study contributes to our understanding on the biodiversity of diaporthalean fungi that are inhabitants of the phyllosphere of trees native to Asia.

## 1. Introduction

The various fungi that inhabit the outer surface and the inner microenvironment of plant leaves are referred to as phyllosphere fungi [1,2]. These fungi include saprophytes, pathogens, epiphytes and endophytes and can affect plant growth and/or help enhance resistances to biotic and abiotic stress [3,4]. There is significant diversity of phyllosphere fungi whose members span diverse phyla in the Fungal Kingdom, particularly within the *Ascomycota* and the *Basidiomycota* [5]. Although studies examining the diversity of fungi in different trophic levels remain limited, fungal diversity on leaves has been shown to be lower than in soil but higher than that found on flowers and fruits, with several studies focusing on entomopathogenic fungi reporting highest diversity in soil, followed by leaves, leaf litter and twigs [6,7]. However, knowledge concerning phyllosphere fungal diversity particularly within the *Diaporthales* remains limited.

The camphor and timber trees, *Cinnamomum camphora* (L.) J. Presl and *Castanopsis fordii* Hance, and the flowering plant, *Schima superba* Gardner & Champ., are widely distributed in south-eastern China, where they can be the dominant members of forest ecosystems [8,9,10]. They play important roles in stabilizing soil, reducing erosion and protecting water sources [11,12]. In addition, these trees have important economic values, with various parts of certain members of the genus *Castanopsis* and *Cinnamomum* frequently employed as part of traditional medicinal practices [11,13]. The crude extracts and chemical constituents derived from *Castanopsis* exhibit a wide range of biological activities, including anti-inflammatory, antioxidant, antimicrobial and other effects [11]. Cinnamic acid, eugenol and cinnamyl alcohol from *Cinnamomum* were the active components of cardiovascular protection [13]. The fungi that associate with *Cinnamomum*, *Castanopsis* and *Schima* plants play different roles, with information concerning fungal *Diaporthales* diversity lacking, especially from diseased leaves [14,15]. Here, we report on the isolation and characterization of new fungal species isolated from diseased leaves of *Cinnamomum camphora*, *Castanopsis fordii* and *Schima superba* in Fujian Province, China. Based on morphological and molecular phylogenetic analyses, we identify three new fungal species, namely *Diaporthe wuyishanensis* sp. nov. in *Diaporthaceae*, *Gnomoniopsis wuyishanensis* sp. nov. in *Gnomoniaceae*, and *Paratubakia schimae* sp. nov. in *Tubakiaceae*. Detailed descriptions and illustrations of the three new species are given. To the best of our knowledge, our data include the first identification and description of *Paratubakia*, (*P. schimae* sp. nov.) in China.

## 2. Materials and Methods

### 2.1. Specimen Sources, Isolation, Morphological Characterization and Selection

Fungal spot disease specimens found on leaves of *Cinnamomum camphora*, *Castanopsis fordii* and *Schima superba* were collected at Meihua Mountain National Nature Reserve, Longyan City and Wuyi Mountain National Nature Reserve, Wuyishan City in Fujian Province, China. The two sampling sites are representative areas of large *Cinnamomum*, *Castanopsis* and *Schima* forests, with high plant diversity, abundant precipitation and more mountains. The leaf specimens were placed in paper bags, which were labeled with the details concerning plant hosts, locations, geographical features and altitudes [16]. Specimens were taken to laboratory and treated as described in Photita et al. [17]. The fungi were isolated using a tissue separation method as follows: ~25 mm^2^ diseased tissue fragments were cut from leaves displaying spot symptoms. The fragments were first sterilized by soaking in 75% ethanol for 60 s and then rinsed once with sterile deionized water for 20 s. Following this, samples were placed in 5% NaOCl for 30 s and then rinsed three times with sterile deionized water for 60 s each time. Finally, fragments were dried on sterilized filter paper and then placed onto potato dextrose agar (PDA) plates for fungal outgrowth [17]. Pure colonies were obtained after sequential passage via culturing of growing fungal colony edges on PDA. Plates were incubated in a light incubator (12:12) at 25 °C. Strains were presumptively identified following three steps: firstly, each strain was sequenced with ITS and *tef1* phylogenetic markers. The BLASTn searches were used to determine the most closely related taxa in the GenBank database with the ITS sequences. Secondly, the family or genus level phylogenetic analysis was conducted using ITS-*tef1* sequences. For *Diaporthe*, strains were assigned to *Diaporthe* different complexes and referred to Dissanayake et al. [18]. Thirdly, amplification of other different loci and phylogenetic analyses were conducted using different multi-locus datasets. Ultimately, herbarium materials were kept at the Herbarium Mycologicum Academiae Sinicae, Institute of Microbiology, Chinese Academy of Sciences, Beijing, China (HMAS), and living cultures were maintained in the China General Microbiological Culture Collection Center (CGMCC). Colony features were imaged with digital camera (Canon EOS 6D MarkII, Tokyo, Japan) at 7 and 15 days after inoculation in indicated media. Microstructures were observed and photographed using a stereo microscope (Nikon SMZ745, Tokyo, Japan) and biological microscope (Ni-U, Tokyo, Japan) with a digital camera (Olympus, Tokyo, Japan) using differential interference contrast (DIC) [19]. Structural measurements were measured using Digimizer 5.4.4 software (https://www.digimizer.com).

### 2.2. DNA Extraction, PCR Amplification and Sequencing

Fungal DNA was directly extracted from growing mycelia on PDA after 5–7 days of growth using the Fungal DNA Mini Kit (OMEGA-D3390, Feiyang Biological Engineering Co., Ltd., Guangzhou, China) according to the product manual. Nucleotide sequences corresponding to seven genetic loci were amplified by polymerase chain reaction (PCR) using a Bio-Rad Thermocycler (Hercules, CA, USA). For *Diaporthe*, the internal transcribed spacer (ITS) region was amplified with primers ITS5 and ITS4, the calmodulin (*cal*) gene with primers CAL-228F and CAL-737R, the histone H3 (*his3*) gene with primers CYLH3F and H3-1b, the translation elongation factor 1-α (*tef1*) gene with primers EF1-728F and TEF1-986R and the beta-tubulin (*tub2*) gene with primers Bt2a and Bt2b as described [20,21,22,23]. The same primers were used for *Gnomoniopsis*, with the exception of amplification of the *tef1* gene, which was performed using primers EF1-728F and EF-2 [20,21,23,24]. For *Paratubakia*, the ITS and *tub2* sequences were amplified with the primers listed above, and in addition, the LSU gene was amplified with primers LROR and LR5, the DNA-directed RNA polymerase II subunit (*rpb2*) gene with primers fRPB2-5F and fRPB2-7cR and the *tef1* gene with primers EF1-728F and EF-2 [20,21,23,24,25,26,27]. The PCR thermal cycle program, primer pairs and sequence are listed in Table 1. The PCR reaction mixture was 25 µL, containing 12.5 μL of 2 × Spark Taq PCR Master Mix (Without Dye) (Shandong Sparkjade Biotechnology Co., Ltd., Jinan, China), 1 μL of template DNA, 1 µL each 10 µM primer (Tsingke, Fuzhou, China) and 9.5 µL of sterile water. Qualified PCR products were checked on electrophoresed in 1% agarose gel (RM19009 and RM02852, ABclonal) and were sequenced by a commercial company (Fuzhou Sunya Biotechnology Co., Ltd., Fuzhou, China).

### 2.3. Sequence Alignment and Phylogenetic Analysis

New sequences generated from this study were deposited in GenBank (Table 2, Table 3 and Table 4). Reference sequences were downloaded from GenBank (Table 2, Table 3 and Table 4). Sequences were aligned with MAFFT v.7 (http://mafft.cbrc.jp/alignment/server/, (accessed on 29 August 2024)) and corrected manually by MEGA 7 software [28,29]. The concatenated aligned sequences were analyzed by Maximum likelihood (ML) and Bayesian inference (BI) methods using the CIPRES Science Gateway portal (https://www.phylo.org/, accessed on (30 August 2024)) and Phylosuite software v. 1.2.3 [30,31]. The Maximum likelihood (ML) analysis was performed with 1000 rapid bootstrap replicates using the GTRGAMMA substitution model by RaxML-HPC2 on ACCESS v. 8.2.12 [32,33]. For Bayesian inference (BI) analyses, Partition Finder2 was used to select the evolutionary model for each locus [34]. Four simultaneous Markov Chain Monte Carlo (MCMC) chains were initiated from random trees with 1 million generations for the *Diaporthe*, 15 million generations for the *Gnomoniopsis* and 5 million generations for the *Tubakiaceae* analyses. In the tests, analyses were sampled every 100 generations. The first 25% of trees were discarded, and remaining trees were used to determine posterior probabilities (PPs). System diagrams were plotted using FigTree v. 1.4.5_pre (https://github.com/rambaut/figtree/releases (accessed on 3 September 2024)).

## 3. Results

### 3.1. Phylogenetic Analyses

For the *Diaporthe virgiliae* species complex, the concatenated sequences dataset for the ITS, *cal*, *his3*, *tef1* and *tub2* genes were analyzed. The alignment included 13 taxa with *Diaporthe shennongjiaensis* as the outgroup (CNUCC 201905) (Figure 1). The sequence dataset contained 2653 characters (*cal*: 1–468, *his3*: 469–936, ITS: 937–1528, *tef1*: 1529–1865, *tub2*: 1866–2653) including gaps. Of these, 2418 characters were constant, 148 variable characters were parsimony-uninformative and 87 characters were parsimony informative. The SYM + I model was proposed for ITS, and the GTR model was proposed for *tef1*, and the HKY + I model was proposed for *cal*, *his3* and *tub2*. The topology of Bayesian analyses was almost identical to the ML tree; thus, the Bayesian tree is shown.

For *Gnomoniopsis* phylogenetic analyses, the concatenated sequence dataset combining the ITS, *tef1* and *tub2* gene loci was used. The alignment included 73 taxa with *Melanconis stilbostoma* as the outgroup (CBS 109778) (Figure 2). The sequence dataset contained 2584 characters (ITS: 1–564, *tef1*: 565–1710, *tub2*: 1711–2584) including gaps. Of these, 1462 characters were constant, 142 variable characters were parsimony-uninformative and 980 characters were parsimony informative. The GTR + I + G model was proposed for ITS, tef1 and tub2 analysis. The topology of the Bayesian analyses was almost identical to the ML tree; thus, the Bayesian tree is shown.

To infer the interspecific relationships between *Paratubakia* within *Tubakiaceae*, a dataset consisting of ITS, LSU, *rpb2*, *tef1* and *tub2* sequences was assembled and analyzed. The alignment included 52 taxa with *Greeneria uvicola* (FI12007) as the outgroup (Figure 3). The sequence dataset contained 3809 characters (ITS: 1–666, LSU: 667–1516, *rpb2*: 1517–2501, *tef1*: 2502–3217, *tub2*: 3218–3809) including gaps. Of these, 2473 characters were constant, 148 variable characters were parsimony-uninformative and 1188 characters were parsimony informative. The GTR + I + G model was proposed for ITS, LSU, *rpb2* and *tub2*, and the HKY + I + G model was proposed for *tef1* analyses. The topology of Bayesian analysis was almost identical to the ML tree; thus, the Bayesian tree is shown.

### 3.2. Taxonomy

#### 3.2.1. *Diaporthe wuyishanensis* W.B. Zhang and J.Z. Qiu, sp. nov., Figure 4

MycoBank Number: MB856022

Etymology: the epithet “*wuyishanensis*” refers to the locality, Wuyi Mountain National Nature Reserve.

Holotype: China: Fujian Province, Wuyi Mountain National Nature Reserve, 27°38′37.88″ N, 117°55′52.39″ E, on diseased leaves of *Cinnamomum camphora*, 7 September 2022, T.C. Mu, holotype HMAS 352949, ex-holotype living culture CGMCC3.27490.

**Figure 4 jof-11-00008-f004:**
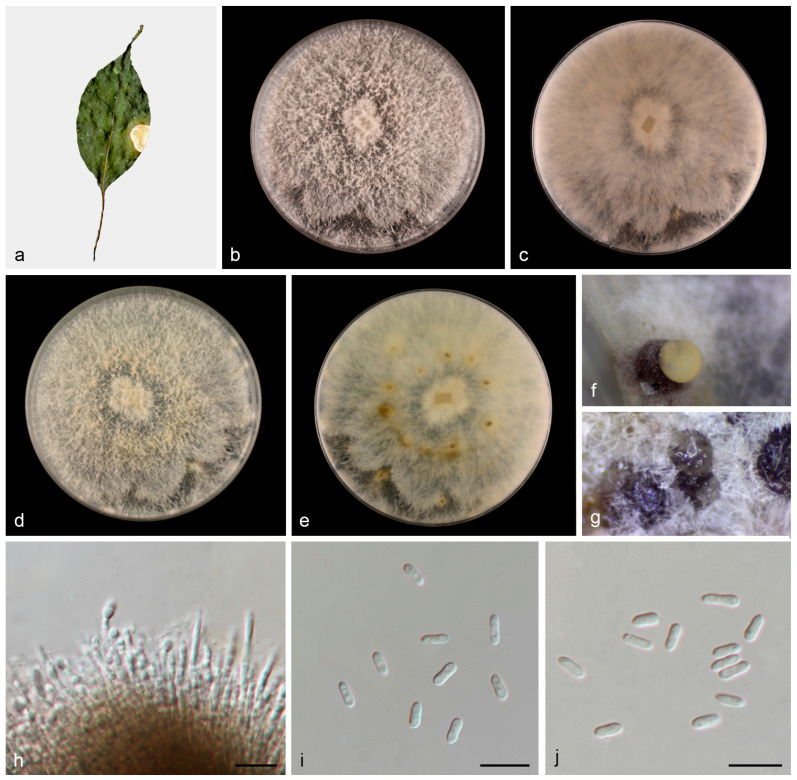
*Diaporthe wuyishanensis* (HMAS 352949). (**a**) Diseased leaves of *Cinnamomum camphora*; (**b**,**c**) surface and reverse sides of colony after 7 days on PDA (**d**,**e**) and 14 days; (**f**,**g**) conidiomata; (**h**) conidiogenous cells and conidia; and (**i**,**j**) alpha conidia. Scale bars: (**h**–**j**) 10 µm.

Description: Asexual morphs: Conidiomata on PDA medium, coriaceous, pycnidial, solitaryor aggregated, superfical, dark brown to black, globose to subglobose, creamy yellowish conidial droplets exuded from ostioles. Conidiophores reduced to conidiogenous cells. Conidiogenous cells hyaline, phialidic, densely aggregated, cylindrical or clavate, straight to slightly curved, 19.7–22.4 × 1.7–2.3 μm, *n* = 20. Conidia septate, hyaline, smooth, cylindrical or subcylindrical, cylindric-clavate, 4.3–6.5 × 1.4–2.6 μm, mean = 5.6 × 1.9 μm, L/W ratio = 3.0, *n* = 30. Beta conidia, gamma conidia and sexual morph not observed.

Culture characteristics: Colonies on PDA were fluffy, and aerial mycelia were abundant, grayish in the center and white at the edge. The surface was initially white, then became pale yellow with age and reverse grayish yellow spots in the middle. Colonies on PDA covered 90 mm plates after 1 week at 25 °C, growth rate 11.8–12.5 mm/day.

Other materials examined: China: Fujian Province, Wuyi Mountain National Nature Reserve, 27°38′37.88″ N, 117°55′52.39″ E, on diseased leaves of *Cinnamomum camphora*, 7 September 2022, T.C. Mu, paratype HMAS 352950, ex-paratype living culture CGMCC3.27491.

Notes: In this study, multigene phylogenetic analysis showed that *Diaporthe wuyishanensis* (CGMCC3.27490 and CGMCC3.27491) formed an independent clade (92% ML/0.99 PP, Figure 1) with *Diaporthe grandiflori* [35]. However, *Diaporthe wuyishanensis* distinguished from *Diaporthe grandiflori* by ITS and *tef1* loci comparison (32/552 in ITS and 19/317 in *tef1*). Morphologically, the alpha conidia of *Diaporthe wuyishanensis* are smaller than *Diaporthe grandiflori* (4.3–6.5 × 1.4–2.6 vs. 6.3–8.3 × 2.8–3.3 μm), and *Diaporthe grandiflori* produces alpha conidia and beta conidia, whereas *Diaporthe wuyishanensis* appears to only produce alpha conidia. Therefore, we introduce this fungus as a new species.

#### 3.2.2. *Gnomoniopsis wuyishanensis* T.C. Mu and J.Z. Qiu, sp. nov., Figure 5

MycoBank Number: MB856023

Etymology: The epithet “*wuyishanensis*” refers to the collection site of the holotype, Wuyi Mountain National Nature Reserve.

Holotype: China: Fujian Province, Wuyi Mountain National Nature Reserve, 27°43′42.05″ N, 117°42′49.13″ E, on diseased leaves of *Castanopsis fordii*, 30 June 2023, T.C. Mu and Z.A. Heng, holotype HMAS 353149, ex-holotype living culture CGMCC3.27836.

**Figure 5 jof-11-00008-f005:**
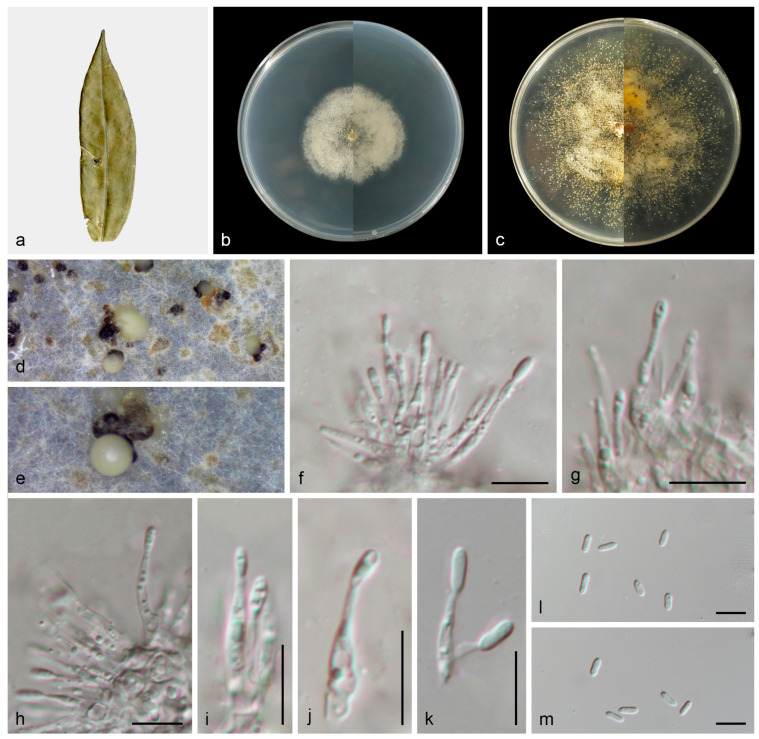
*Gnomoniopsis wuyishanensis* (HMAS 353149). (**a**) Diseased leaves of *Castanopsis fordii*; (**b**) surface and reverse sides of colony after 7 days on PDA (**c**) and 14 days; (**d**,**e**) conidiomata; (**f**–**k**) conidiogenous cells and conidia; and (**l**,**m**) conidia. Scale bars: (**f**–**m**) 10 µm.

Description: Asexual morphs: Conidiomata developed on PDA, pycnidial, solitary, black, creamy conidial droplets exuded from ostioles. Conidiophores are indistinct, frequently reduced to conidiogenous cells. Conidiogenous cells hyaline, smooth, phialidic, clavate, straight to sinuous, attenuate towards apex, 10.8–23.8 × 1.4–2.6 μm, *n* = 20. Conidia hyaline, smooth, aseptate, oblong to ellipsoid, subcylindrical, 5.1–8.7 × 1.4–3.0 μm, mean = 6.6 ×2.1 μm, L/W ratio = 3.2, *n* = 30. Sexual morph not observed.

Culture characteristics: Colonies flat with irregular margin, white aerial mycelium then becoming pale gray by age. PDA attaining 36.0–40.7 mm in diameter after 1 week at 25 °C, growth rate 5.1–5.8 mm/day. PDA attaining 77.2–80.1 mm in diameter after 2 weeks at 25 °C, growth rate 5.5–5.7 mm/day.

Other materials examined: China: Fujian Province, Wuyi Mountain National Nature Reserve, 27°43′42.05″ N, 117°42′49.13″ E, on diseased leaves of *Castanopsis fordii*, 30 June 2023, T.C. Mu and Z.A. Heng, paratype HMAS 353148, ex-paratype living culture CGMCC3.27834.

Notes: *Gnomoniopsis wuyishanensis* was isolated from diseased leaves of *C. fordii* from Fujian Province, China. *Gnomoniopsis fagacearum* (CFCC 54414) was collected from diseased leaves of *Castanopsis eyrei* from Fujian Province, China, by Jiang et al. [30]. Phylogenetically, *Gnomoniopsis wuyishanensis* forms a well-supported (83% ML/1 PP, Figure 2) clade that is close, but distinct from *Gnomoniopsis guangdongensis*, *Gnomoniopsis lithocarpi* and *Gnomoniopsis silvicola* [30,36]. Morphologically, the conidia of *Gnomoniopsis wuyishanensis* are larger than *Gnomoniopsis guangdongensis*, *Gnomoniopsis lithocarpi* and *Gnomoniopsis silvicola* (5.1–8.7 × 1.4–3.0 vs. 4.3–5.2 × 1.4–2.0 vs. 4.0–5.8 × 1.7–2.4 vs. 4.3–5.9 × 1.9–2.7 μm). Therefore, we introduce this taxon as a new species.

#### 3.2.3. *Paratubakia schimae* T.C. Mu and J.Z. Qiu, sp. nov., Figure 6

MycoBank Number: MB856024

Etymology: The epithet “*schimae*” refers to the genus of the host plant on which it was collected, *Schima*.

Holotype: China: Fujian Province, Longyan City, Meihua Mountain National Nature Reserve, 25°39′20.91″ N, 116°55′32.01″ E, on diseased leaves of *Schima superba*, 14 September 2022, J.H. Chen and C.J. Yang, holotype HMAS 353150, ex-holotype living culture CGMCC3.27842.

Description: Asexual morphs: Conidiomata formed on the surface of PDA medium, pycnothyria grouped together, initially white and apricot, then became black with age. Conidiophores reduced to conidiogenous cells. Conidiogenous cells, hyaline to pale brown, smooth, thin-walled, phialidic, obclavate, 13.0–20.5 × 4.4–5.8 μm, *n* = 20. Conidia hyaline to slightly pigmented, smooth, solitary, globose to subglobose, with inconspicuous to conspicuous hilum, 10.6–13.7 × 9.4–11.8 μm, mean = 12.1 × 10.5 μm, L/W ratio = 1.2, *n* = 30. Sexual morph not observed.

Culture characteristics: Colonies flat with regular margin, aerial mycelium white. This species can produce red pigment during growth, which causes the surface and reverse sides of PDA medium to change from colorless to red. PDA attaining 37.7–40.3 mm in diameter after 1 week at 25 °C, growth rate 5.4–5.8 mm/day. PDA attaining 78.7–85.6 mm in diameter after 2 weeks at 25 °C, growth rate 5.6–6.1 mm/day.

Other materials examined: China: Fujian Province, Longyan City, Meihua Mountain National Nature Reserve, 25°39′20.91″ N, 116°55′32.01″ E, on diseased leaves of *Schima superba*, 14 September 2022, J.H. Chen and C.J. Yang, paratype HMAS 353151, ex-paratype living culture CGMCC3.27855.

Notes: In this study, *Paratubakia schimae* form a well-supported (96% ML/1 PP, Figure 3) clade that is close to *Paratubakia subglobosa* within *Tubakiaceae* trees. However, *Paratubakia schimae* is distinguished from *Paratubakia subglobosa* by ITS, *tef1*, *tub2* and *rpb2* loci (24/519 in ITS, 38/569 in *tef1*, 19/498 in *tub2* and 20/760 in *rpb2*) [37]. Morphologically, the conidiogenous cells of *Paratubakia schimae* are larger than *Paratubakia subglobosa* (13.0–20.5 × 4.4–5.8 vs. 8.0–12.0 × 2.0–3.0 μm). Therefore, we introduce this fungus as a new species.

**Figure 6 jof-11-00008-f006:**
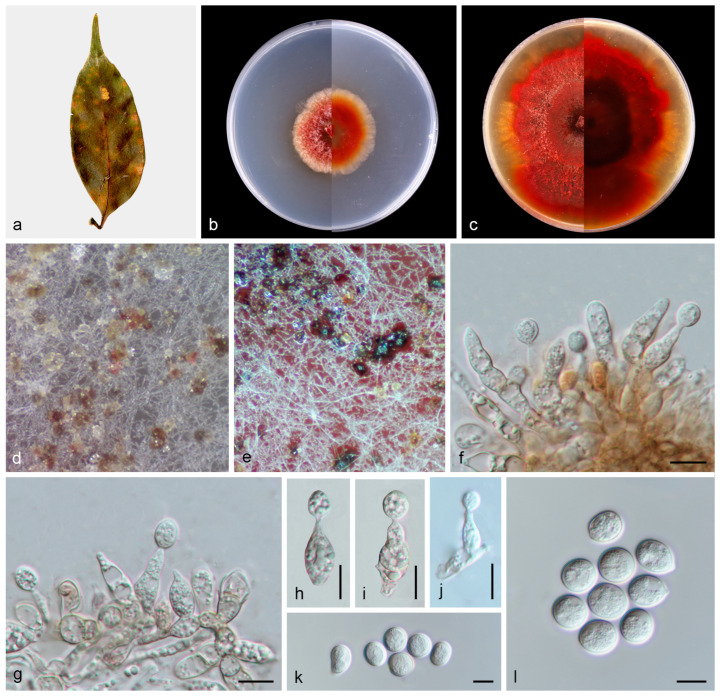
*Paratubakia schimae* (HMAS 353150). (**a**) Diseased leaves of *Schima superba*; (**b**) surface and reverse sides of colony after 7 days on PDA (**c**) and 14 days; (**d**,**e**) conidiomata; (**f**–**j**) conidiogenous cells and conidia; and (**k**,**l**) conidia. Scale bars: (**f**–**l**) 10 µm.

## 4. Discussion

Studies examining fungal diversity on leaves have led to the discovery of a variety of new species and taxa that include pathogens as well as potentially beneficial epi- and endophytes [38,39,40]. *Diaporthales* Nannf. (phylum *Ascomycota*) constitutes an important order of phyllosphere fungi [41,42]. Recent advances include the description of a new family, *Pyrisporaceae* C.M. Tian & N. Jiang, erected based on the type genus *Pyrispora* C.M. Tian & N. Jiang, with *Pyrispora castaneae* as the type species, which was collected from leaves of the Chinese chestnut (*Castanea mollissima*) [43]. Also, *Obovoideisporodochium* Z. X. Zhang, J. W. Xia & X. G. Zhang was established with the type species *Obovoideisporodochium lithocarpi* isolated from leaves of *Lithocarpus fohaiensis* [44]. In a survey of fungi associated with plant leaves in south-western China, eight new species of *Diaporthe* were identified from tea (*Camellia sinensis*): *Castanea mollissima*, *Chrysalidocarpus lutescens*, *Elaeagnus conferta*, *Elaeagnus pungens*, *Heliconia metallica*, *Heterostemma grandiflorum*, *Litchi chinensis*, *Machilus pingii*, *Melastoma malabathricum* and *Millettia reticulate* [35].

The genus *Diaporthe* Nitschke (syn. *Phomopsis* (Sacc.) Bubák) belongs to *Diaporthaceae* Höhn. ex Wehm. (*Diaporthales*), with *Diaporthe eres* as the type species [45,46]. Species of *Diaporthe* include both plant pathogens and endophytes, typically with broad host ranges, as well as saprophytes [47]. Currently, more 1200 epithets of *Diaporthe* and 983 of *Phomopsis* have been recorded in the Index Fungorum (http://www.indexfungorum.org/; (accessed 10 September 2024)). Based on five single genetic loci, as well as multigene phylogentic analyses, the genus *Diaporthe* was re-structured with seven sections proposed: Betulicola, Crotalariae, Eres, Foeniculina, Psoraleae-pinnatae, Rudis and Sojae, with boundaries for 13 species and 15 species complexes [18]. The lengthy phylogenetic trees of the entire *Diaporthe* and data analysis were avoided, which provides mycologist and taxonomists with the convenience of focusing on specific sections, species complexes and species. Here, a new species, *Diaporthe wuyishanensis*, is introduced into the *Diaporthe virgiliae* species complex, based on both morphological differences and multi-locus (ITS, *cal*, *his3*, *tef1* and *tub2*) molecular analyses. The *Diaporthe virgiliae* species complex contains five species, viz. *Diaporthe grandiflori*, *Diaporthe heterophyllae*, *Diaporthe penetriteum*, *Diaporthe virgiliae* and *Diaporthe zaofenghuang*, previously [18].

*Gnomoniopsis* Berl. is a genus in the *Gnomoniaceae* G. Winter (*Diaporthales*) with *Gnomoniopsis chamaemori* as the type species [48]. *Gnomoniopsis* was originally studied as a subgenus within *Gnomonia* Ces. & De Not. because of their similar morphology [49]. However, *Gnomoniopsis* has been separated from *Gnomonia* by means of morphology, phylogeny and host associations [36,48,49]. As important pathogens of agricultural and forestry trees, flowers and fruit, species of *Gnomoniopsis* can cause signficant plant damage and resultant economic losses [50,51,52,53]. It is reported that leaf spot diseases of oak (*Quercus alba* and *Quercus rubra*) have also been caused by *Gnomoniopsis clavulata* infection in North America [54], and *Gnomoniopsis fragariae* is reported to result in leaf blotch disease of strawberry in Europe [52]. Within the past 10–12 years, *Gnomoniopsis smithogilvyi* has been isolated from diseased chestnut in Spain, Portugal and Greece, which are important chestnut-producing countries in Europe [55,56]. *Gnomoniopsis castaneae* infection damages the fruit of chestnuts and can cause cankers and necrosis on leaves. Cankers have also been reported on chestnut wood, red oak and hazelnut trees and are currently considered major threats to global chestnut production, potentially threatening the reintroduction of American chestnut, as the fungus has been found in North America [57].

With respect to host plants, *Gnomoniopsis* appears to mainly inhabit three plant families, viz. *Rosaceae*, *Fagaceae* and *Onagraceae* [49,58]. This is reflected in phylogram plant host analyses, in which *Gnomoniopsis* divides into three clades: Rosaceousclade, Fagaceousclade and Onagraceousclade, with most species currently assigned within the former two clades. Species of *Gnomoniopsis* have host-specific features in each clade, although the molecular basis for this specificity remains unknown. The new species we report, *Gnomoniopsis wuyishanensis*, fits within the Fagaceous clade. These characterizations and placements allow for surveillance of potential outbreaks in relevant hosts.

The genus *Paratubakia* U. Braun & C. Nakash. belongs to *Tubakiaceae* U. Braun, J.Z. Groenew. & Crous (*Diaporthales*) with *Paratubakia subglobosa* as the type species [37]. Based on morphological and phylogenetic analyses, *Phaeotubakia* (type species: *Phaeotubakia lithocarpicola*) has more recently been proposed. Based on a multigene phylogeny (LSU and *rpb2*), *Paratubakia subglobosa* and *Paratubakia subglobosoides* have been shown to form an independent branch of *Tubakiaceae*. Currently, *Paratubakia* includes only two species: *Paratubakia subglobosa* and *Paratubakia subglobosoides*. *Tubakiaceae* has been proposed to accommodate the genera *Apiognomonioides*, *Involutscutellula*, *Oblongisporothyrium*, *Paratubakia*, *Racheliella*, *Saprothyrium*, *Sphaerosporithyrium* and *Tubakia* based on LSU sequence alignment and type genus *Tubakia* [37]. Subsequently, *Obovoideisporodochium* was established based on the type species *Obovoideisporodochium lithocarpi* [44], and *Ellipsoidisporodochium* was erected based on the type species *Ellipsoidisporodochium photiniae* [59]. Both *Obovoideisporodochium lithocarpi* and *Phaeotubakia lithocarpicola* and most *Tubakiaceae* species were found from *Fagaceae* plants [60]. Species of *Paratubakia* were only found and described from the Japanese blue oak (*Quercus glauca*) [37]. Here, we report on a new species *Paratubakia schimae*, with, to the best of our knowledge, the genus found and described in China for the first time. As this is the first description of *Paratubakia* in China, its distribution and potential host range remain unknown. However, our data suggest significant likelihood for additional discovery.

## 5. Conclusions

In this study, based on morphological features and multigene phylogenetic analyses, we described three new species of *Diaporthales* distributed within three different genera from China, viz. *Diaporthe wuyishanensis*, *Gnomoniopsis wuyishanensis* and *Paratubakia schimae*. These studies reveal a high diversity of phyllosphere fungi and help plant pathologists, taxonomists and phytologists to improve understanding of plant–fungal interactions.

## Figures and Tables

**Figure 1 jof-11-00008-f001:**
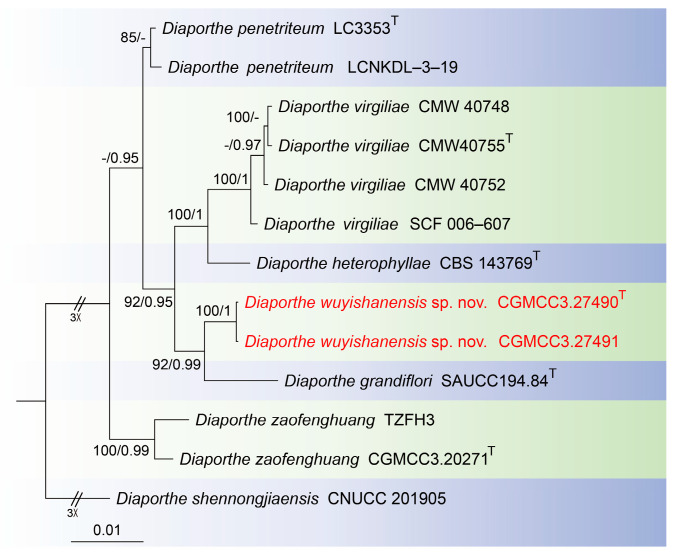
Consensus tree of *Diaporthe virgiliae* species complex inferred from Bayesian inference analyses based on the combined ITS, *cal, his3, tef1* and *tub2* sequence dataset, with *Diaporthe shennongjiaensis* (CNUCC 201905) as the outgroup. The Maximum likelihood (ML) bootstrap support values and Bayesian posterior probabilities (BPPs) above 80% and 0.90 are shown at the nodes. Strains marked with “T” are ex-type, ex-epitype and ex-neotype. The isolates from this study are indicated in red.

**Figure 2 jof-11-00008-f002:**
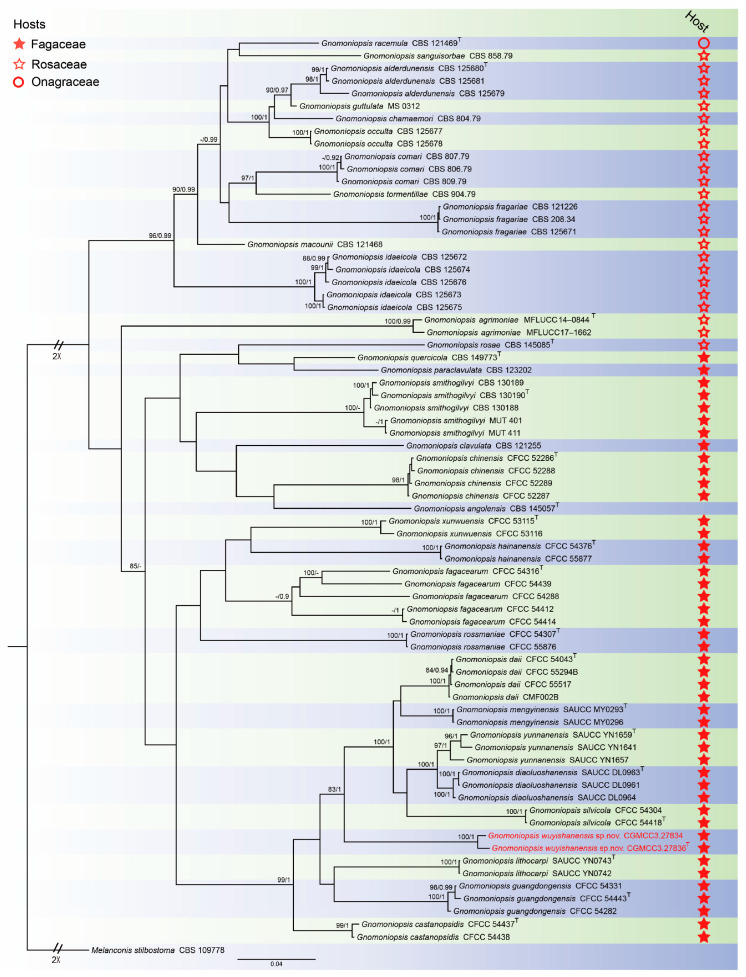
Consensus tree of *Gnomoniopsis* inferred from Bayesian inference analyses based on the combined ITS, *tef1* and *tub2* sequence dataset, with *Melanconis stilbostoma* (CBS 109778) as the outgroup. The Maximum likelihood (ML) bootstrap support values and Bayesian posterior probabilities (BPPs) above 80% and 0.90 were shown at the nodes. Strains marked with “T” are ex-type, ex-epitype and ex-neotype. The isolates from this study are indicated in red.

**Figure 3 jof-11-00008-f003:**
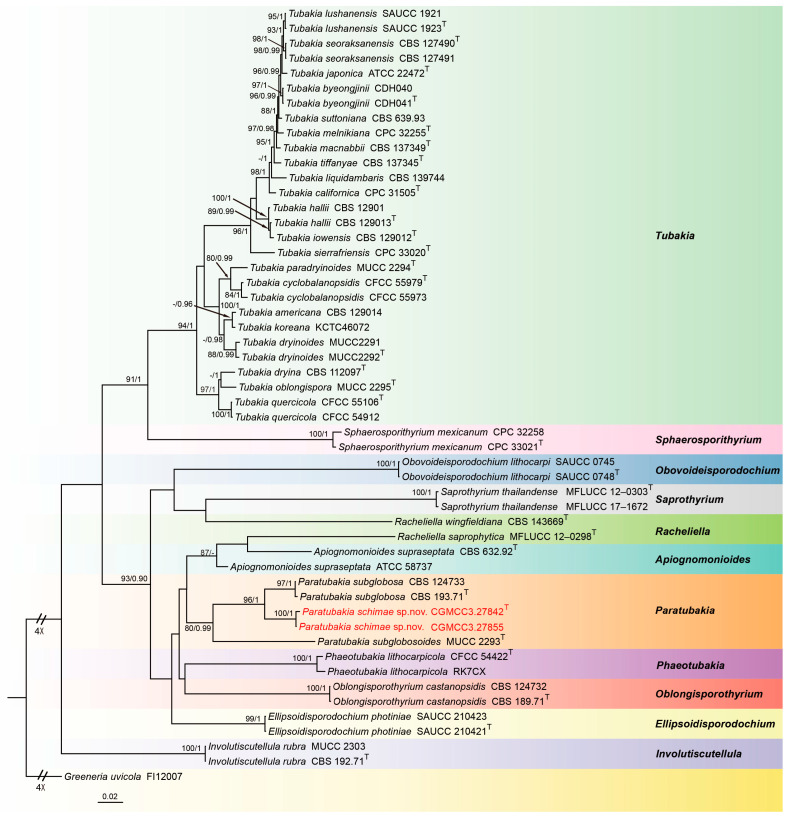
Consensus tree of *Tubakiaceae* inferred from Bayesian inference analyses based on the combined ITS, LSU, *rpb2*, *tef1* and *tub2* sequence dataset, with *Greeneria uvicola* (FI12007) as the outgroup. The Maximum likelihood (ML) bootstrap support values and Bayesian posterior probabilities (BPPs) above 80% and 0.90 were shown at the nodes. Strains marked with “T” are ex-type, ex-epitype and ex-neotype. The isolates from this study are indicated in red.

**Table 1 jof-11-00008-t001:** Target sequences, primer pairs and PCR programs used for application in this study.

Loci	PCR Primers	Sequence (5′–3′)	PCR Cycles	References
ITS	ITS5	GGA AGT AAA AGT CGT AAC AAG G	(95 °C: 30 s, 55 °C: 30 s, 72 °C: 1 min) × 35 cycles	[20]
	ITS4	TCC TCC GCT TAT TGA TAT GC		
LSU	LROR	GTA CCC GCT GAA CTT AAG C	(95 °C: 30 s, 52 °C: 30 s, 72 °C: 1 min) × 35 cycles	[25,26]
	LR5	TCC TGA GGG AAA CTT CG		
*cal*	CAL-228F	GAG TTC AAG GAG GCC TTC TCC C	(95 °C: 30 s, 54 °C: 30 s, 72 °C: 1 min) × 35 cycles	[21]
	CAL-737R	CAT CTT CTG GCC ATC ATG G		
*rpb2*	fRPB2-5F	GAY GAY MGW GAT CAY TTY GG	(95 °C: 30 s, 56 °C: 30 s, 72 °C: 1 min) × 35 cycles	[27]
	fRPB2-7cR	CCC ATW GCY TGC TTM CCC AT		
*his3*	CYLH3F	AGG TCC ACT GGT GGC AAG	(95 °C: 30 s, 58 °C: 30 s, 72 °C: 1 min) × 35 cycles	[22,23]
	H3-1b	GCG GGC GAG CTG GAT GTC CTT		
*tef1*	EF1-728F	CAT CGA GAA GTT CGA GAA GG	(95 °C: 30 s, 48 °C: 30 s, 72 °C: 1 min) × 35 cycles	[21,24]
	EF-2	GGA RGT ACC AGT SAT CAT GTT		
	EF1-728F	CAT CGA GAA GTT CGA GAA GG	(95 °C: 30 s, 52 °C: 30 s, 72 °C: 1 min) × 35 cycles	[21]
	TEF1-986R	TAC TTG AAG GAA CCC TTA CC		
*tub2*	Bt2a	GGT AAC CAA ATC GGT GCT GCT TTC	(95 °C: 30 s, 53 °C: 30 s, 72 °C: 1 min) × 35 cycles	[23]
	Bt2b	ACC CTC AGT GTA GTG ACC CTT GGC		

**Table 2 jof-11-00008-t002:** Information of specimens and GenBank accession numbers of the sequences used in the analysis of the *Diaporthe virgiliae* species complex.

Species	Culture/Voucher	Host/Substrate	Locations	GenBank Accession Number				
				ITS	*tub2*	*tef1*	*cal*	*his3*
*Diaporthe wuyishanensis* sp. nov.	CGMCC3.27490 *	*Cinnamomum camphora*	China	PQ385851	PQ404036	PQ404034	PQ404030	PQ404032
*Diaporthe wuyishanensis* sp. nov.	CGMCC3.27491	*Cinnamomum camphora*	China	PQ385852	PQ404037	PQ404035	PQ404031	PQ404033
*Diaporthe grandiflori*	SAUCC194.84 *	*Heterostemma grandiflorum*	China	MT822612	MT855809	MT855924	MT855691	MT855580
*Diaporthe heterophyllae*	CBS 143769 *	*Acacia heterophylla*	France	MG600222	MG600226	MG600224	MG600218	MG600220
*Diaporthe penetriteum*	LC3353 *	*Camellia sinensis*	China	KP714505	KP714529	KP714517	–	KP714493
*Diaporthe penetriteum*	NKDL–3–19	*Citrus sinensis*	China	MW202992	MW208612	MW221578	MW221746	MW221677
*Diaporthe shennongjiaensis*	CNUCC 201905	*Juglans regia*	China	MN216229	MN227012	MN224672	MN224551	MN224560
*Diaporthe virgiliae*	CMW 40748	*Virgilia oroboides*	South Africa	KP247566	KP247575	–	–	–
*Diaporthe virgiliae*	CMW 40752	*Virgilia oroboides*	South Africa	KP247570	KP247579	–	–	–
*Diaporthe virgiliae*	CMW40755 *	*Virgilia oroboides*	South Africa	KP247573	KP247582	–	–	–
*Diaporthe virgiliae*	SCF 006–607	*Cyclopia subternata*	South Africa	MW959685	MW979256	MT833892	–	MT833906
*Diaporthe zaofenghuang*	CGMCC3.20271 *	*Prunus persica*	China	MW477883	MW480875	MW480871	MW480867	MW480863
*Diaporthe zaofenghuang*	TZFH3	*Prunus persica*	China	MW477884	MW480876	MW480872	MW480868	MW480864

Notes: Newly generated sequences are in bold. The ex-type, ex-epitype and ex-neotype strains are marked with *.

**Table 3 jof-11-00008-t003:** Information of specimens and GenBank accession numbers of the sequences used in the analysis of *Gnomoniopsis*.

Species	Culture/Voucher	Host/Substrate	Locations	GenBank Accession Number		
				ITS	*tef1*	*tub2*
*Gnomoniopsis agrimoniae*	MFLUCC 14–0844 *	*Agrimonia eupatoria*	Italy	–	MF377585	*–*
*Gnomoniopsis agrimoniae*	MFLUCC 17–1662	*Agrimonia eupatoria*	Italy	–	MF377586	*–*
*Gnomoniopsis alderdunensis*	CBS 125680 *	*Rubus parviflorus*	USA	GU320825	GU320801	GU320787
*Gnomoniopsis alderdunensis*	CBS 125681	*Rubus parviflorus*	USA	GU320827	GU320802	GU320789
*Gnomoniopsis alderdunensis*	CBS 125679	*Rubus pedatus*	USA	GU320826	GU320813	GU320788
*Gnomoniopsis angolensis*	CPC 33595 = CBS 145057 *	unknown host plant	Angola	MK047428	–	–
*Gnomoniopsis castanopsidis*	CFCC 54437 *	*Castanopsis hystrix*	China	MZ902909	MZ936385	–
*Gnomoniopsis castanopsidis*	CFCC 54438	*Castanopsis hystrix*	China	MZ902910	MZ936386	–
*Gnomoniopsis chamaemori*	CBS 804.79	*Rubus chamaemorus*	Finland	GU320817	GU320809	GU320777
*Gnomoniopsis chinensis*	CFCC 52286 *	*Castanea mollissima*	China	MG866032	MH545370	MH545366
*Gnomoniopsis chinensis*	CFCC 52288	*Castanea mollissima*	China	MG866034	MH545372	MH545368
*Gnomoniopsis chinensis*	CFCC 52287	*Castanea mollissima*	China	MG866033	MH545371	MH545367
*Gnomoniopsis chinensis*	CFCC 52289	*Castanea mollissima*	China	MG866035	MH545373	MH545369
*Gnomoniopsis clavulata*	CBS 121255	*Quercus falcata*	USA	EU254818	EU221934	EU219211
*Gnomoniopsis comari*	CBS 807.79	*Comarum palustre*	Finland	EU254822	GU320814	GU320779
*Gnomoniopsis comari*	CBS 809.79	*Comarum palustre*	Switzerland	EU254823	GU320794	GU320778
*Gnomoniopsis comari*	CBS 806.79	*Oryza sativa*	UK	EU254821	GU320810	EU219156
*Gnomoniopsis daii*	CFCC 54043 *	*Castanea mollissima*	China	MZ902911	MZ936387	MZ936403
*Gnomoniopsis daii*	CFCC 55517	*Castanea mollissima*	China	MN598671	MN605519	MN605517
*Gnomoniopsis daii*	CMF002B	*Castanea mollissima*	China	MN598672	MN605520	MN605518
*Gnomoniopsis daii*	CFCC 55294B	*Quercus aliena*	China	MZ902912	MZ936388	MZ936404
*Gnomoniopsis diaoluoshanensis*	SAUCC DL0963 *	*Castanopsis chinensis*	China	ON753744	ON759769	ON759777
*Gnomoniopsis diaoluoshanensis*	SAUCC DL0964	*Castanopsis chinensis*	China	ON753743	ON759768	ON759776
*Gnomoniopsis diaoluoshanensis*	SAUCC DL0961	*Castanopsis chinensis*	China	ON753745	ON759770	ON759778
*Gnomoniopsis fagacearum*	CFCC 54412	*Castanopsis chunii*	China	MZ902917	MZ936393	MZ936409
*Gnomoniopsis fagacearum*	CFCC 54414	*Castanopsis eyrei*	China	MZ902915	MZ936391	MZ936407
*Gnomoniopsis fagacearum*	CFCC 54288	*Castanopsis faberi*	China	MZ902913	MZ936389	MZ936405
*Gnomoniopsis fagacearum*	CFCC 54316 *	*Lithocarpus glaber*	China	MZ902916	MZ936392	MZ936408
*Gnomoniopsis fagacearum*	CFCC 54439	*Quercus variabilis*	China	MZ902914	MZ936390	MZ936406
*Gnomoniopsis fragariae = Gnomoniopsis fructicola*	CBS 121226	*Fragaria vesca*	USA	EU254824	EU221961	EU219144
*Gnomoniopsis fragariae = Gnomoniopsis fructicola*	CBS 125671	*Fragaria* sp.	USA	GU320816	GU320793	GU320776
*Gnomoniopsis fragariae = Gnomoniopsis fructicola*	CBS 208.34	*Fragaria* sp.	USA	EU254826	EU221968	EU219149
*Gnomoniopsis guangdongensis*	CFCC 54443 *	*Castanopsis fargesii*	China	MZ902918	MZ936394	MZ936410
*Gnomoniopsis guangdongensis*	CFCC 54331	*Castanopsis fargesii*	China	MZ902919	MZ936395	MZ936411
*Gnomoniopsis guangdongensis*	CFCC 54282	*Castanopsis fargesii*	China	MZ902920	MZ936396	MZ936412
*Gnomoniopsis guttulata*	MS 0312	*Agrimonia eupatoria*	Bulgaria	EU254812	–	–
*Gnomoniopsis hainanensis*	CFCC 54376 *	*Castanopsis hainanensis*	China	MZ902921	MZ936397	MZ936413
*Gnomoniopsis hainanensis*	CFCC 55877	*Castanopsis hainanensis*	China	MZ902922	MZ936398	MZ936414
*Gnomoniopsis idaeicola*	CBS 125673	*Rubus pedatus*	USA	GU320824	GU320798	GU320782
*Gnomoniopsis idaeicola*	CBS 125675	*Rubus pedatus*	USA	GU320822	GU320799	GU320783
*Gnomoniopsis idaeicola*	CBS 125676	*Rubus pedatus*	USA	GU320821	GU320811	GU320784
*Gnomoniopsis idaeicola*	CBS 125672	*Rubus* sp.	USA	GU320823	GU320797	GU320781
*Gnomoniopsis idaeicola*	CBS 125674	*Rubus* sp.	France	GU320820	GU320796	GU320780
*Gnomoniopsis lithocarpi*	SAUCC YN0743*	*Lithocarpus fohaiensis*	China	ON753749	ON759765	ON759783
*Gnomoniopsis lithocarpi*	SAUCC YN0742	*Lithocarpus fohaiensis*	China	ON753750	ON759764	ON759782
*Gnomoniopsis macounii*	CBS 121468	*Spiraea* sp.	USA	EU254762	EU221979	EU219126
*Gnomoniopsis mengyinensis*	SAUCC MY0293 *	*Castanea mollissima*	China	ON753741	ON759766	ON759774
*Gnomoniopsis mengyinensis*	SAUCC MY0296	*Castanea mollissima*	China	ON753742	ON759767	ON759775
*Gnomoniopsis occulta*	CBS 125677	*Potentilla* sp.	USA	GU320828	GU320812	GU320785
*Gnomoniopsis occulta*	CBS 125678	*Potentilla* sp.	USA	GU320829	GU320800	GU320786
*Gnomoniopsis paraclavulata*	CBS 123202	*Quercus alba*	USA	GU320830	GU320815	GU320775
*Gnomoniopsis quercicola*	IRAN 4313C = CBS 149773 *	*Quercus brantii*	Iran	OR540614	OR561996	OR561907
*Gnomoniopsis racemula*	CBS 121469 *	*Triticum aestivum*	USA	EU254841	EU221889	EU219125
*Gnomoniopsis rosae*	CPC 34440 = CBS 145085 *	*Rosa* sp.	New Zealand	MK047451	–	–
*Gnomoniopsis rossmaniae*	CFCC 54307 *	*Castanopsis hainanensis*	China	MZ902923	MZ936399	MZ936415
*Gnomoniopsis rossmaniae*	CFCC 55876	*Castanopsis hainanensis*	China	MZ902924	MZ936400	MZ936416
*Gnomoniopsis sanguisorbae*	CBS 858.79	*Sanguisorba minor*	Switzerland	GU320818	GU320805	GU320790
*Gnomoniopsis silvicola*	CFCC 54304	*Castanopsis hystrix*	China	MZ902925	MZ936401	MZ936417
*Gnomoniopsis silvicola*	CFCC 54418 *	*Quercus serrata*	China	MZ902926	MZ936402	MZ936418
*Gnomoniopsis smithogilvyi*	MUT 401	*Castanea sativa*	Italy	HM142946	KR072537	KR072532
*Gnomoniopsis smithogilvyi*	MUT 411	*Castanea sativa*	New Zealand	HM142948	KR072538	KR072533
*Gnomoniopsis smithogilvyi*	CBS 130190 *	*Castanea* sp.	Australia	JQ910642	JQ910645	JQ910639
*Gnomoniopsis smithogilvyi*	CBS 130189	*Castanea* sp.	Australia	JQ910644	JQ910647	JQ910641
*Gnomoniopsis smithogilvyi*	CBS 130188	*Castanea* sp.	Australia	JQ910643	KR072536	JQ910640
*Gnomoniopsis tormentillae*	CBS 904.79	*Potentilla* sp.	Switzerland	EU254856	GU320795	EU219165
*Gnomoniopsis wuyishanensis* sp. nov.	CGMCC3.27834	*Castanopsis fordii*	China	PQ381256	PQ404016	PQ404018
*Gnomoniopsis wuyishanensis* sp. nov.	CGMCC3.27836 *	*Castanopsis fordii*	China	PQ381257	PQ404017	PQ404019
*Gnomoniopsis xunwuensis*	CFCC 53115 *	*Castanopsis fissa*	China	MK432667	MK578141	MK578067
*Gnomoniopsis xunwuensis*	CFCC 53116	*Castanopsis fissa*	China	MK432668	MK578142	MK578068
*Gnomoniopsis yunnanensis*	SAUCC YN1659 *	*Castanea mollissima*	China	ON753746	ON759771	ON759779
*Gnomoniopsis yunnanensis*	SAUCC YN1657	*Castanea mollissima*	China	ON753747	ON759772	ON759780
*Gnomoniopsis yunnanensis*	SAUCC YN1641	*Castanea mollissima*	China	ON753748	ON759773	ON759781
*Melanconis stilbostoma*	CBS 109778	*Betula pendula*	Australia	DQ323524	EU221886	EU219104

Notes: Newly generated sequences are in bold. The ex-type, ex-epitype and ex-neotype strains are marked with *.

**Table 4 jof-11-00008-t004:** Information of specimens and GenBank accession numbers of the sequences used in the analysis of *Tubakiaceae*.

Species	Culture/Voucher	Host/Substrate	Locations	GenBank Accession Number				
				ITS	LSU	*tef1*	*tub2*	*rpb2*
*Apiognomonioides supraseptata*	CBS 632.92 *	*Quercus glauca*	Japan	MG976447	MG976448	–	–	–
*Apiognomonioides supraseptata*	ATCC 58737	*Quercus glauca*	Japan	–	AF277127	–	–	–
*Ellipsoidisporodochium photiniae*	SAUCC 210421 *	*Photinia serratifolia*	China	OK175559	OK189532	OK206440	OK206442	OK206438
*Ellipsoidisporodochium photiniae*	SAUCC 210423	*Photinia serratifolia*	China	OK175560	OK189533	OK206441	OK206443	OK206439
*Greeneria uvicola*	FI12007	unknown host plant	Uruguay	HQ586009	GQ870619	–	–	–
*Involutiscutellula rubra*	CBS 192.71 *	*Quercus phillyraeoides*	Japan	MG591899	MG591993	MG592086	MG592180	MG976476
*Involutiscutellula rubra*	MUCC 2303	*Quercus phillyraeoides*	Japan	MG591900	MG591994	MG592087	MG592181	MG976477
*Oblongisporothyrium castanopsidis*	CBS 124732	*Castanopsis cuspidata*	Japan	MG591849	MG591942	MG592037	MG592131	MG976453
*Oblongisporothyrium castanopsidis*	CBS 189.71 *	*Castanopsis cuspidata*	Japan	MG591850	MG591943	MG592038	MG592132	MG976454
*Obovoideisporodochium lithocarpi*	SAUCC 0748 *	*Lithocarpus fohaiensis*	China	MW820279	MW821346	MZ996876	MZ962157	MZ962155
*Obovoideisporodochium lithocarpi*	SAUCC 0745	*Lithocarpus fohaiensis*	China	MW820280	MW821347	MZ996877	MZ962158	MZ962156
*Paratubakia schimae* sp. nov.	CGMCC3.27842 *	*Schima superba*	China	PQ408642	PQ408644	PQ404020	PQ404022	PQ404024
*Paratubakia schimae* sp. nov.	CGMCC3.27855	*Schima superba*	China	PQ408643	PQ408645	PQ404021	PQ404023	PQ404025
*Paratubakia subglobosa*	CBS 124733	*Quercus glauca*	Japan	MG591913	MG592008	MG592102	MG592194	MG976489
*Paratubakia subglobosa*	CBS 193.71 *	*Quercus glauca*	Japan	MG591914	MG592009	MG592103	MG592195	MG976490
*Paratubakia subglobosoides*	MUCC 2293 *	*Quercus glauca*	Japan	MG591915	MG592010	MG592104	MG592196	MG976491
*Phaeotubakia lithocarpicola*	CFCC 54422 *	*Lithocarpus glaber*	China	OP951017	OP951015	OQ127584	OQ127586	–
*Phaeotubakia lithocarpicola*	RK7CX	*Lithocarpus glaber*	China	OP951018	OP951016	OQ127585	OQ127587	–
*Racheliella saprophytica*	MFLUCC 12–0298 *	*Syzygium cumini*	Thailand	KJ021933	KJ021935	–	–	–
*Racheliella wingfieldiana*	CBS 143669 *	*Syzigium guineense*	South Africa	MG591911	MG592006	MG592100	MG592192	MG976487
*Saprothyrium thailandense*	MFLUCC 12–0303 *	Decaying leaf	Thailand	MF190163	MF190110	–	–	–
*Saprothyrium thailandense*	MFLUCC 17–1672	Decaying leaf	Thailand	MF190164	MF190111	–	–	–
*Sphaerosporithyrium mexicanum*	CPC 32258	*Quercus eduardi*	Mexico	MG591895	MG591989	MG592082	MG592176	–
*Sphaerosporithyrium mexicanum*	CPC 33021 *	*Quercus eduardi*	Mexico	MG591896	MG591990	MG592083	MG592177	MG976473
*Tubakia americana*	CBS 129014	*Quercus macrocarpa*	USA	MG591873	MG591966	MG592058	MG592152	MG976449
*Tubakia byeongjinii*	CDH040	*Quercus variabilis*	Republic of Korea	OR727896	OR727898	OR732731	–	–
*Tubakia byeongjinii*	CDH041 *	*Quercus variabilis*	Republic of Korea	OR727897	OR727899	OR732732	–	–
*Tubakia californica*	CPC 31505 *	*Quercus kelloggii*	USA	MG591835	MG591928	MG592023	MG592117	MG976451
*Tubakia cyclobalanopsidis*	CFCC 55979 *	*Quercus glauca*	China	OP114639	–	OP254247	OP329290	–
*Tubakia cyclobalanopsidis*	CFCC 55973	*Quercus glauca*	China	OP114640	–	OP254248	OP329291	–
*Tubakia dryina*	CBS 112097 *	*Quercus robur*	Italy	MG591851	MG591944	MG592039	MG592133	MG976455
*Tubakia dryinoides*	MUCC2291	*Castanea crenata*	Japan	MG591877	MG591969	MG592062	MG592156	MG976460
*Tubakia dryinoides*	MUCC2292 *	*Quercus phillyraeoides*	Japan	MG591878	MG591970	MG592063	MG592157	MG976461
*Tubakia hallii*	CBS 129013 *	*Quercus stellata*	USA	MG591880	MG591972	MG592065	MG592159	MG976462
*Tubakia hallii*	CBS 12901	*Quercus stellata*	USA	MG591881	MG591973	MG592066	MG592160	–
*Tubakia iowensis*	CBS 129012 *	*Quercus macrocarpa*	USA	MG591879	MG591971	MG592064	MG592158	–
*Tubakia japonica*	ATCC 22472 *	*Castanea crenata*	Japan	MG591886	MG591978	MG592071	MG592165	MG976465
*Tubakia koreana*	KCTC46072	*Quercus mongolica*	Republic of Korea	KP886837	–	–	–	–
*Tubakia liquidambaris*	CBS 139744	*Liquidambar styraciflua*	USA	MG605068	MG605077	MG603578	–	–
*Tubakia lushanensis*	SAUCC 1921	*Quercus palustris*	China	MW784677	MW784850	MW842262	MW842265	MW842268
*Tubakia lushanensis*	SAUCC 1923 *	*Quercus palustris*	China	MW784678	MW784851	MW842261	MW842264	MW842267
*Tubakia macnabbii*	CBS 137349 *	*Quercus palustris*	USA	MG605069	–	MG603579	–	–
*Tubakia melnikiana*	CPC 32255 *	*Quercus canbyi*	Mexico	MG591893	MG591987	MG592080	MG592174	MG976472
*Tubakia oblongispora*	MUCC 2295 *	*Quercus serrata*	Japan	MG591897	MG591991	MG592084	MG592178	MG976474
*Tubakia paradryinoides*	MUCC 2294 *	*Quercus acutissima*	Japan	MG591898	MG591992	MG592085	MG592179	MG976475
*Tubakia quercicola*	CFCC 55106 *	*Quercus aliena* var. *acuteserrata*	China	OP114635	–	OP254243	OP254289	–
*Tubakia quercicola*	CFCC 54912	*Quercus aliena* var. *acuteserrata*	China	OP114636	–	OP254244	OP254290	–
*Tubakia seoraksanensis*	CBS 127490 *	*Quercus mongolica*	Republic of Korea	MG591907	KP260499	MG592094	MG592186	–
*Tubakia seoraksanensis*	CBS 127491	*Quercus mongolica*	Republic of Korea	HM991735	KP260500	MG592095	MG592187	MG976484
*Tubakia sierrafriensis*	CPC 33020 *	*Quercus eduardi*	Mexico	MG591910	MG592005	MG592099	MG592191	MG976486
*Tubakia suttoniana*	CBS 639.93	*Quercus* sp.	Italy	MG591921	MG592016	MG592110	MG592202	MG976493
*Tubakia tiffanyae*	CBS 137345 *	*Quercus rubra*	USA	MG605081	–	MG603581	–	–

Notes: Newly generated sequences are in bold. The ex-type, ex-epitype and ex-neotype strains are marked with *.

## Data Availability

All sequences generated in this study were submitted to the NCBI database.

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
