# Peer review of "New Species of *Diaporthales* (*Ascomycota*) from Diseased Leaves in Fujian Province, China"

_jof, 2024, doi:10.3390/jof11010008_

Round 1

Reviewer 1 Report

Three new species of Diaporthales, from diseased leaves in Fujian Province, China were intoduced based on multigene phylogenetic analyses and morphological characters. This research will contribute to the understanding of Diaporthales phyllosphere fungal species diversity.

1. The formating need to be improved. Please check the MS text format carefully and correct. Such as: A space need to be added before "[**]" in the MS text.  

2. For new species, depositing duplicates of the herbarium (holotype) and culture (ex-holotype) are do suggested. Isotype and ex-isotype are requested. 

3. For Diaporthe camphorae, according to the plate (Fig. 4, h), conidiophores probably are reduced to conidiogenous cells, Just describing conidiogenous cells is enough. The photo of conidiogenous cells (h) is not clear, authors can observe more photos to clarify if they are holoblastic, enteroblastic or phialidic.

4. For Paratubakia schimae, according to Fig.6, f-j, conidiophores are reduced to conidiogenous cells. Conidiogenous cells are phialidic, and more than 10 μm in lenfgth. The descriptions of conidiophores and conidiogenous cells need to be improved. The number of conidiogenous cells measured should be added in description.

More comments and corrections, please see the attachment. 

Author Response

Dear Editors and Reviewers:

Thank you for your letter and comments relating to our manuscript entitled “New Species of Sac Fungi (Diaporthales) from Diseased Leaves in Fujian Province, China” (ID: jof-3307771). The comments were very helpful in revising and improving our manuscript as well as emphasizing the significance to our research. We have read the comments carefully and made corrections accordingly. Revised portions are marked in blue in the manuscript. The main corrections in the paper and our responses to the reviewer’s comments are given below. We appreciate your consideration of this manuscript for publication in the Journal of Fungi.

Responses to the comments of the reviewer:

Reviewer#1

Comments 1: What was the technique you used to isolate fungi?

Response 1: A tissue separation method was used to isolate fungi.

Fungal specimens were isolated as follow: ~25 mm2 diseased tissues fragments were cut from leaves displaying spot symptoms. The fragments were first sterilized by being soaked in a 75% ethanol solution for 60 s. Fragments werethen rinsed once with sterile deionized water for 20 s and then placed in 5% NaOCl for 30 s. Plant fragments were subsequently rinsed three times with sterile deionized water for 60 s each time. Finally, samples were dried on sterilized filter paper. The dry fragments were then placed onto PDA plates.

Comments 2: What was your criterion?

Response 2: We refer to the sequences used in published articles about Diaporthe, Gnomoniopsis and Paratubakia and use websites (Index Fungorum and MycoBank) to check and supplement the species involved in the sequences.

Comments 3: The formating need to be improved. Please check the MS text format carefully and correct. Such as: A space need to be added before "[**]" in the MS text.

Response 3: We have revised it.

Comments 4: For new species, depositing duplicates of the herbarium (holotype) and culture (ex-holotype) are do suggested. Isotype and ex-isotype are requested.

Response 4: Thank you very much for your suggestions! We have sent the duplicates of the herbarium (holotype) and culture (ex-holotype), isotype and ex-isotype to Herbarium Mycologicum Academiae Sinicae, Institute of Microbiology, Chinese Academy of Sciences, Beijing, China (HMAS) and China General Microbiological Culture Collection Center (CGMCC).

Comments 5: What's their texture like? membranous, coriaceous or carbonaceous? According to f, g the conidiomata appear to be membranous.

Response 5: Conidiomata is coriaceous.

Comments 6: For Diaporthe camphorae, according to the plate (Fig.4,h), conidiophores probably are reduced to conidiogenous cells, Just describing conidiogenous cells is enough. The photo of conidiogenous cells (h) is not clear, authors can observe more photos to clarify if they are holoblastic, enteroblastic or phialidic.

Response 6: Based on our observation of more photos about Diaporthe camphorae, we confirm that conidiophores reduced to conidiogenous cells. Conidiogenous cells are phialidic.

Conidiophores reduced to conidiogenous cells. Conidiophores cells hyaline, phialidic, densely aggregated, cylindrical or clavate, straight to slightly curved, 19.7–22.4 × 1.7–2.3 μm, n = 20.

Comments 7: For Paratubakia schimae, according to Fig. 6, f-j, conidiophores are reduced to conidiogenous cells. Conidiogenous cells are phialidic, and more than 10 μm in length. The descriptions of conidiophores and conidiogenous cells need to be improved. The number of conidiogenous cells measured should be added in description.

Response 7: About 20 conidiogenous cells were measured.

Conidiophores reduced to conidiogenous cells. Conidiophores cells, hyaline to pale brown, smooth, thin-walled, phialidic, obclavate, 13.0–20.5 × 4.4–5.8 μm, n = 20.

We tried our best to improve the manuscript and made some changes marked in blue in revised paper which will not influence the content and framework of the paper. We appreciate for Editors/Reviewers’ warm work earnestly and hope the revision will meet with your approval. Once again, thank you very much for your comments and suggestions.

Kind regards,

Junzhi Qiu

E-mail address: junzhiqiu@126.com

Reviewer 2 Report

The article is valuable because it describes 3 new species of ascomycete fungi of the order Diaporthales: Diaporthe camphorae (from camphor leaves), Gnomoniopsis wuyishanensis (from Castanopsis fordii leaves), Paratubakia schimae (from Schima superba leaves). However, the manuscript cannot be published in its current format. Firstly, the authors clearly state the objective of the work, which should be to describe certain potentially pathogenic fungi (something they do not demonstrate experimentally) in native trees of the Asian flora, especially in China. Thus, in the introduction they should tell us about the ecological and human importance of these trees, as well as the current problems of the phytopathogenic species of fungi that infect these plants, given that they have collected leaves with lesions caused by these fungi. The authors' treatment of the phylloplane fungi is also not correct, because they only process the material to isolate those that apparently show phytopathogenic activity, leaving no record of those other saprobic species. The references in the Introduction to the knowledge of the genera Diaporthe, Gnomoniopsis and Paratubakia should be moved to a Discussion section, since it cannot be assumed without prior knowledge that the authors will find these taxa in their samples. The technique for isolating the fungi, beyond being referenced (Photita et al.), should be briefly described. According to the Global Biodiversity Information Facility (https://www.gbif.org), the current name of the camphor tree is Cinnamomum camphora (L.) J. Presl, not as cited by the authors (Camphora officinarum). The authors should highlight why they have chosen natural areas as collection sites, and not urban parks or other environments, and should highlight their climatological, geological and phytogeographic characteristics in the Materials and Methods section. The authors also do not mention how they have presumptively identified their isolates, whether by molecular techniques using the sequences of the different phylogenetic markers by performing a search using the BLAST algorithm for the most similar sequences in the corresponding database, or by using phenotypic techniques, or both combined. The authors do not mention what criteria they have used in the selection of the sequences of the molecular markers of interest corresponding to the species included in their phylogenetic trees. For example, the article by Huang S, Xia J, Zhang X, and Sun W (2021. Morphological and phylogenetic analyses reveal three new species of Diaporthe from Yunnan, China. MycoKeys 78: 49–77. https://doi.org/10.3897/mycokeys.78.60878), describing three new species for the genus Diaporthe, includes sequences from 115 strains of species of that genus, while the authors of the present work include sequences from 11 strains belonging to 6 species, in addition to those corresponding to two strains of the proposed new species. Regarding the description of the new taxa, these contain some errors (e.g., conidiophores cells instead conidiogenous cells).

The article is valuable because it describes 3 new species of ascomycete fungi of the order Diaporthales: Diaporthe camphorae (from camphor leaves), Gnomoniopsis wuyishanensis (from Castanopsis fordii leaves), Paratubakia schimae (from Schima superba leaves). However, the manuscript cannot be published in its current format. Firstly, the authors clearly state the objective of the work, which should be to describe certain potentially pathogenic fungi (something they do not demonstrate experimentally) in native trees of the Asian flora, especially in China. Thus, in the introduction they should tell us about the ecological and human importance of these trees, as well as the current problems of the phytopathogenic species of fungi that infect these plants, given that they have collected leaves with lesions caused by these fungi. The authors' treatment of the phylloplane fungi is also not correct, because they only process the material to isolate those that apparently show phytopathogenic activity, leaving no record of those other saprobic species. The references in the Introduction to the knowledge of the genera Diaporthe, Gnomoniopsis and Paratubakia should be moved to a Discussion section, since it cannot be assumed without prior knowledge that the authors will find these taxa in their samples. The technique for isolating the fungi, beyond being referenced (Photita et al.), should be briefly described. According to the Global Biodiversity Information Facility (https://www.gbif.org), the current name of the camphor tree is Cinnamomum camphora (L.) J. Presl, not as cited by the authors (Camphora officinarum). The authors should highlight why they have chosen natural areas as collection sites, and not urban parks or other environments, and should highlight their climatological, geological and phytogeographic characteristics in the Materials and Methods section. The authors also do not mention how they have presumptively identified their isolates, whether by molecular techniques using the sequences of the different phylogenetic markers by performing a search using the BLAST algorithm for the most similar sequences in the corresponding database, or by using phenotypic techniques, or both combined. The authors do not mention what criteria they have used in the selection of the sequences of the molecular markers of interest corresponding to the species included in their phylogenetic trees. For example, the article by Huang S, Xia J, Zhang X, and Sun W (2021. Morphological and phylogenetic analyses reveal three new species of Diaporthe from Yunnan, China. MycoKeys 78: 49–77. https://doi.org/10.3897/mycokeys.78.60878), describing three new species for the genus Diaporthe, includes sequences from 115 strains of species of that genus, while the authors of the present work include sequences from 11 strains belonging to 6 species, in addition to those corresponding to two strains of the proposed new species. Regarding the description of the new taxa, these contain some errors (e.g., conidiophores cells instead conidiogenous cells). Other comments in the attached file.

Author Response

Dear Editors and Reviewers:

Thank you for your letter and comments relating to our manuscript entitled “New Species of Sac Fungi (Diaporthales) from Diseased Leaves in Fujian Province, China” (ID: jof-3307771). The comments were very helpful in revising and improving our manuscript as well as emphasizing the significance to our research. We have read the comments carefully and made corrections accordingly. Revised portions are marked in blue in the manuscript. The main corrections in the paper and our responses to the reviewer’s comments are given below. We appreciate your consideration of this manuscript for publication in the Journal of Fungi.

Responses to the comments of the reviewer:

Reviewer#2

Comments 1: The authors clearly state the objective of the work, which should be to describe certain potentially pathogenic fungi (something they do not demonstrate experimentally) innative trees of the Asian flora, especially in China. Thus, in the introduction they should tell us about the ecological and human importance of these trees, as well as the current problems of the phytopathogenic species of fungi that infect these plants, given that they have collected leaves with lesions caused by these fungi.

Response 1: Camphora, Castanopsis and Schimaare widely distributed in south-eastern China. They play important roles in stabilizing soil, reducing soil erosion, and protecting water sources. Even certain genus (Castanopsis) is frequently employed as traditional medicinal resources. Fungi exist in different parts of Camphora, Castanopsis and Schima plants and play different roles. However, Diaporthales diversity of the phyllosphere is rarely reported, especially from diseased leaves.

Comments 2: The authors' treatment of the phylloplane fungi is also not correct, because they only process the material to isolate those that apparently show phytopathogenic activity, leaving no record of those other saprobic species.

Response 2: In this study, we take diseased plant leaves as the research objects. The fungi were isolated using the tissue separation methodas described in Photita et al. In future research, we will try to find saprobic species and record.

Comments 3: The references in the Introduction to the knowledge of the genera DiaportheGnomoniopsis and Paratubakia should be moved to a Discussion section, since it cannot be assumed without prior knowledge that the authors will find these taxa in their samples.

Response 3: We have revised it.

Fungal diversity on leaves has led to the discovery of a variety of new species and taxa. Diaporthales Nannf. (phylum Ascomycota) constitutes of important order of phyllosphere fungi. Recent advances include the description of a new family, Pyrisporaceae C.M. Tian & N. Jiang, erected based on the type genus Pyrispora C.M. Tian & N. Jiang, with Pyrispora castaneae as the type species, which was collected from leaves of the Chinese chestnut (Castanea mollissima). Also, Obovoideisporodochium Z. X. Zhang, J. W. Xia & X. G. Zhang was established with the type species Obovoideisporodochium lithocarpi isolated from leaves of Lithocarpus fohaiensis. In a survey of fungi associated with plant leaves in south-western China, eight new species of Diaporthewere identified from tea (Camellia sinensis), Castanea mollissima, Chrysalidocarpus lutescens, Elaeagnus conferta, Elaeagnus pungens, Heliconia metallica, Heterostemma grandiflorum, Litchi chinensis, Machilus pingii, Melastoma malabathricum, Millettia reticulata.

The genus Diaporthe Nitschke (syn. Phomopsis (Sacc.) Bubák) belongs to Diaporthaceae Höhn. ex Wehm. (Diaporthales), with Diaporthe eres as the type species. Species of Diaporthe include both plant pathogens and endophytes, typically with broad host ranges, as well as saprophytes. Every year between 2011 and 2024, new species of Diaporthe were published, even though more 1290 epithets of Diaporthe and 983 of Phomopsis have already been recorded in Index Fungorum (http://www.indexfungorum.org/; accessed 10 September 2024). Based on five single gene phylogenies and multigene phylogeny, the genus Diaporthe was re-structured with 7 sections proposed: Betulicola, Crotalariae, Eres, Foeniculina, Psoraleae-pinnatae, Rudis and Sojae, with boundaries for 13 species and 15 species-complexes. The lengthy phylogenetic trees of the entire Diaporthe and data analysis were avoided which provides mycologist and taxonomists with the convenience of focusing on specific section, species-complexes and species. In the current study, a new species, Diaporthe camphorae, was introduced into the Diaporthe virgiliae species complex, based on morphology coupled to multilocus (ITS, cal, his3, tef1 and tub2) molecular analyses. The Diaporthe virgiliae species complex contains 5 species, viz. Diaporthe grandiflori, Diaporthe heterophyllae, Diaporthe penetriteum, Diaporthe virgiliae and Diaporthe zaofenghuang previously. As the number of Diaporthe specimens collected increases, the number and range of section, species-complexes and species may also increase and expand.

Gnomoniopsis Berl. is a genus in the Gnomoniaceae G. Winter (Diaporthales) with Gnomoniopsis chamaemori as the type species. Gnomoniopsis was originally studied as a subgenus within Gnomonia Ces. & De Not. because of their similar morphology. However, Gnomoniopsis has been separated from Gnomonia by means morphology, phylogeny, and host associations. As important pathogens of agricultural and forestry trees, flowers, and fruit, species of Gnomoniopsis can cause signficant plant damage and resultant economic losses. It is reported that leaf spot diseases of oak (Quercus alba and Quercus rubra) have also been caused by Gnomoniopsis clavulata infection in North America, and Gnomoniopsis fragariae is reported to result in leaf blotch disease of strawberry in Europe. From 2012 to 2024, Gnomoniopsis smithogilvyi has been isolated from diseased chestnut in Spain, Portugal and Greece which are important chestnut producing countries in Europe. Gnomoniopsis castaneae infection damages the fruit of chestnuts and can cause cankers and necrosis on leaves. Cankers have also been reported on chestnut wood, red oak, and hazelnut trees, and are currently considered major threats to global chestnut production, potentially threatening the reintroduction of American chestnut, as the fungus has been found in North America.

From the perspective of the host plant, it has been found that Gnomoniopsis inhabits three plant families, viz. Rosaceae, Fagaceae and Onagraceae. Based on phylogram analyses and plant-host, Gnomoniopsis can be divided into three clades: Rosaceous clade, Fagaceous clade and Onagraceous clade, with most species currently assigned within the former two clades. Species of Gnomoniopsis have host-specific features in each clade, although the molecular basis for this specificity remains unknown. The new species we report, Gnomoniopsis wuyishanensis fits within the Fagaceous clade. This also provides direction for the prevention of related plant diseases.

The genus Paratubakia U. Braun & C. Nakash. belongs to Tubakiaceae U. Braun, J.Z. Groenew. & Crous (Diaporthales) with Paratubakia subglobosa as the type species. Based on morphological and phylogenetic analyses, Phaeotubakia (type species: Phaeotubakia lithocarpicola) has proposed more recently been proposed. Based on a multigene phylogeny (LSU and rpb2), Paratubakia subglobosa and Paratubakia subglobosoides have been shown to form an independent branch of Tubakiaceae. Currently, Paratubakia includes only two species: Paratubakia subglobosa and Paratubakia subglobosoides. Tubakiaceae has been proposed to accommodate the genera Apiognomonioides, Involutscutellula, Oblongisporothyrium, Paratubakia, Racheliella, Saprothyrium, Sphaerosporithyrium, Tubakiabased on LSU sequence alignment and type genus Tubakia. Subsequently, Obovoideisporodochium was established based on the type species Obovoideisporodochium lithocarpi, and Ellipsoidisporodochium was erected based on the type species Ellipsoidisporodochium photiniae. Both Obovoideisporodochium lithocarpi and Phaeotubakia lithocarpicola and most Tubakiaceae species were found from Fagaceae plants [58]. Species of Paratubakia were only found and described from the Japanese blue oak (Quercus glauca). Here we report on a new species Paratubakia schimae, with, to the best of our knowledge, the genus foundand described in China for the first time. As this is the first description of Paratubakia in China, its distribution and potential host range remains unknown. However, our data suggest significant likelihood for additional discovery.

Comments 4: The technique for isolating the fungi, beyond being referenced (Photita et al.), should be briefly described.

Response 4: We have revised it.

The fungi were isolated using the tissue separation method. About ~25 mm2 diseased tissues fragments were cut from leaves with apparent spot symptoms. The fragments were first sterilized by being soaked in a 75% ethanol solution for 60 s. After that, they were rinsed once with sterile deionized water for 20 s. Following this, they were moved into a 5% NaOCl for 30s. Then, they were rinsed three times with sterile deionized water for 60 s each time. Finally, fragments were dried on sterilized filter paper. The dry fragments were then placed onto new PDA plates

Comments 5: According to the Global Biodiversity Information Facility (https://www.gbif.org), the current name of the camphor tree is Cinnamomum camphora (L.) J. Presl, not as cited by the authors (Camphora officinarum).

Response 5: This plant of scientific name has undergone some changes in 2022. Based on phylogenetic analyses, Yang et al. proposed Cinnamomum is not monophyletic and Cinnamomum should be divided into two genera, i.e., Cinnamomum and Camphora. “Cinnamomum camphora” was change to “Camphora officinarum”. We use the latest scientific names in this article.

Yang, Z.; Liu, B.; Yang, Y.; Ferguson, D.K. Phylogeny and Taxonomy of Cinnamomum (Lauraceae). Ecology and Evolution 2022, 12, 10, e9378.

Comments 6: The authors should highlight why they have chosen natural areas as collection sites, and not urban parks or other environments, and should highlight their climatological, geological and phytogeographic characteristics in the Materials and Methods section.

Response 6: We have revised it.

Fungal spot diseased leaf specimen of Camphora officinarum, Castanopsis fordii and Schima superba were collected at Meihua Mountain National Nature, Longyan City and Wuyi Mountain National Nature Reserve Reserve, Wuyishan City in Fujian Province, China. The two sampling sites are representative areas of Camphora, Castanopsis and Schima plants, with high plant diversity, abundant precipitation and more mountains.

Comments 7: The authors also do not mention how they have presumptively identified their isolates, whether by molecular techniques using the sequences of the different phylogenetic markers by performing a search using the BLAST algorithm for the most similar sequences in the corresponding database, or by using phenotypic techniques, or both combined.

Response 7: Strains were presumptively identified following two steps: firstly, each strain will be cultured and its colony morphology and other phenotype characterization will be observed. Similar phenotypes will be grouped together. Secondly, each strain will be sequenced with ITS and tef1 phylogenetic markers. The BLAST algorithm was used to search for the most similar sequences in the NCBI database.

Comments 8: The authors do not mention what criteria they have used in the selection of the sequences of the molecular markers of interest corresponding to the species included in their phylogenetic trees. For example, the article by Huang S, Xia J, Zhang X, and Sun W (2021. Morphological and phylogenetic analyses reveal three new species of Diaporthe from Yunnan, China. MycoKeys 78: 49–77. https://doi.org/10.3897/mycokeys.78.60878), describing three new species for the genus Diaporthe, includes sequences from 115 strains of species of that genus, while the authors of the present work include sequences from 11 strains belonging to 6 species, in addition to those corresponding to two strains of the proposed new species.

Response 8: For Diaporthe, the phylogenetic analysis was conducted using ITS-tef1 sequences, and the Diaporthe strains were assigned to Diaporthe different sections and complexes. For Gnomoniopsis and Paratubakia, amplification of different loci refers to published articles.

Comments 9: Regarding the description of the new taxa, these contain some errors (e.g., conidiophores cells instead conidiogenous cells).

Response 9: We have revised it.

We tried our best to improve the manuscript and made some changes marked in blue in revised paper which will not influence the content and framework of the paper. We appreciate for Editors/Reviewers’ warm work earnestly and hope the revision will meet with your approval. Once again, thank you very much for your comments and suggestions.

Kind regards,

Junzhi Qiu

E-mail address: junzhiqiu@126.com

Reviewer 3 Report

The paper is an important contribution on the knowledge of phyllosphere fungi. The paper is well contsructed and the research was conducted properly.

The submitted paper New Species of Sac Fungi (Diaporthales) from Diseased Leaves in Fujian Province, China " by X. Guan and colleagues is interesting and it is well constructed. It is a new addition of new species into the Diaporthales.

I think we can and should adopt the “rule” that all scientific names of all taxa ranks can be in italics. See it here: Thines et al. IMA Fungus (2020) 11:25; https://doi.org/10.1186/s43008-020-00048-6

The paper is clear, well written and well organised. 

Conclusions could be more developed. Please adrees to my detail comments on this.

Please, see my detail comments.

The submitted paper New Species of Sac Fungi (Diaporthales) from Diseased Leaves in Fujian Province, China " by X. Guan and colleagues is interesting and it is well constructed. It is a new addition of new species into the Diaporthales.

I think we can adopt the “rule” that all scientific names of all taxa ranks can be in italics. See it here: Thines et al. IMA Fungus (2020) 11:25; https://doi.org/10.1186/s43008-020-00048-6

The paper is clear, well written and well organised.

The tittle was well chosen!

The abstract is clear, pointing out the main outcomes of the article. Also, keywords were well chosen.

The Introduction of the paper is well organised, and the objectives/outcomes are clearly indicated.

The Materials and Methods section is well organized, the methods are adequately described and with detail. It is accompanied by proper tables! Also, all the adequate tools were employed by the authors!

I just would only place Figure 1 after the tittle of the subsection of Results appears, after subsection 3.2. It is within the previous subsection of Materials and Methods…

Results are illustrated with the proper figures and images of good quality. Phylogenetic analyses were well performed, and the phylogenetic trees are clear and properly constructed.

Taxonomy is also clear. And the descriptions and nomenclature were deposited in Mycobank.

Nevertheless, in figure 4 I would change the colour of the letters “f-g-h” to white, instead of black. The same for letter “d” in figure 6.

Discussion is well constructed. If possible, a section of Conclusions could be created. Or alternatively, the authors should develop more the last period of conclusions (at the very end of Discussion), highlighting the hosts where the new fungal species were found, to relate these data with potential economic and ecological impacts. Nothing or little is said about the importance of the host tress (particularly in China) where the new fungal species were found. Some comments on this could be useful (this can be added for example, also after this paragraph (lines 362-363) as a new one… Or in the “Conclusions”.

These studies revealed a high diversity of phyllosphere fungi and help plant pathologists, taxonomists and phytologist to improve understanding of plant-fungus interactions.”

In page 19, in the line 360 “…[52]. In this study, three diverse new specieis”, correct the word “species”.

Please insert a dot in line 397:

“This also provides direction for the prevention of related plant diseases”

The list of references is suitable to me.

Author Response

Dear Editors and Reviewers:

Thank you for your letter and comments relating to our manuscript entitled “New Species of Sac Fungi (Diaporthales) from Diseased Leaves in Fujian Province, China” (ID: jof-3307771). The comments were very helpful in revising and improving our manuscript as well as emphasizing the significance to our research. We have read the comments carefully and made corrections accordingly. Revised portions are marked in blue in the manuscript. The main corrections in the paper and our responses to the reviewer’s comments are given below. We appreciate your consideration of this manuscript for publication in the Journal of Fungi.

Responses to the comments of the reviewer:

Reviewer#3

Comments 1: I think we can adopt the “rule” that all scientific names of all taxa ranks can be in italics. See it here: Thines et al. IMA Fungus (2020) 11:25; https://doi.org/10.1186/s43008-020-00048-6

Response 1: We have revised it.

Comments 2: In the Materials and Methods section, I just would only place Figure 1 after the tittle of the subsection of Results appears, after subsection 3.2. It is within the previous subsection of Materials and Methods…

Response 2: We have made some adjustments to make it easier for readers to read.

Comments 3: In the Results section, in figure 4 I would change the colour of the letters “f-g-h” to white, instead of black. The same for letter “d” in figure 6.

Response 3: We have changed them.

Comments 4: If possible, a section of Conclusions could be created. Or alternatively, the authors should develop more the last period of conclusions (at the very end of Discussion), highlighting the hosts where the new fungal species were found, to relate these data with potential economic and ecological impacts. Nothing or little is said about the importance of the host tress (particularly in China) where the new fungal species were found. Some comments on this could be useful (this can be added for example, also after this paragraph (lines 362-363) as a new one… Or in the “Conclusions”.“These studies revealed a high diversity of phyllosphere fungi and help plant pathologists, taxonomists and phytologist to improve understanding of plant-fungus interactions.”

Response 4: Thank you very much for your suggestions!

Conclusions

In this study, based on morphological features and multigene phylogenetic analyses, we describe three new species of Diaporthales distributed within three different genera from China, viz. Diaporthe camphorae, Gnomoniopsis wuyishanensis and Paratubakia schimae. These studies revealed a high diversity of phyllosphere fungi and help plant pathologists, taxonomists and phytologist to improve understanding of plant-fungus interactions.

Comments 5: In page 19, in the line 360 “…[52]. In this study, three diverse new specieis”, correct the word “species”.

Response 5: We have revised it.

Comments 6: Please insert a dot in line 397: “This also provides direction for the prevention of related plant diseases”. The list of references is suitable to me.

Response 6: We have revised it.

We tried our best to improve the manuscript and made some changes marked in blue in revised paper which will not influence the content and framework of the paper. We appreciate for Editors/Reviewers’ warm work earnestly and hope the revision will meet with your approval. Once again, thank you very much for your comments and suggestions.

Kind regards,

Junzhi Qiu

E-mail address: junzhiqiu@126.com

Reviewer 4 Report

The authors achieved what they wanted to share. The manuscript is relevant and important to taxonomy, phylogeny and morphology. My main concern is about the discussion. For example, Diaporthe there are several species, despite they focus on the main groups, they should make a strong discussion. It seems to be simple for a genus with so many species.

“fromtea” change to from tea

Quercus rubr” change to Quercus rubra

The phyllogenetic tree is shown in the methodology, not in the results. The authors must put on the correct called of the text

3.3. Taxonomy

Section: Description

“solitaryor” change to “solitary or”

“Orsubcylindrical” change to or subcylindrical

“toonly” separate these words

Cellshyaline separate these words

“Morphologically, the conidiophores of Paratubakia schimae are large than Paratubakia subglobosa” change to “ are largeR than”

Author Response

Dear Editors and Reviewers:

Thank you for your letter and comments relating to our manuscript entitled “New Species of Sac Fungi (Diaporthales) from Diseased Leaves in Fujian Province, China” (ID: jof-3307771). The comments were very helpful in revising and improving our manuscript as well as emphasizing the significance to our research. We have read the comments carefully and made corrections accordingly. Revised portions are marked in blue in the manuscript. The main corrections in the paper and our responses to the reviewer’s comments are given below. We appreciate your consideration of this manuscript for publication in the Journal of Fungi.

Responses to the comments of the reviewer:

Reviewer#4

Comments 1: My main concern is about the discussion. For example, Diaporthe there are several species, despite they focus on the main groups, they should make a strong discussion. It seems to be simple for a genus with so many species.

Response 1: We have revised it.

Every year between 2011 and 2024, new species of Diaporthe were published, even though more 1290 epithets of Diaporthe and 983 of Phomopsis have already been recorded in Index Fungorum (http://www.indexfungorum.org/; accessed 10 September 2024). Based on five single gene phylogenies and multigene phylogeny, the genus Diaporthe was re-structured with 7 sections proposed: Betulicola, Crotalariae, Eres, Foeniculina, Psoraleae-pinnatae, Rudis and Sojae, with boundaries for 13 species and 15 species-complexes. The lengthy phylogenetic trees of the entire Diaporthe and data analysis were avoided which provides mycologist and taxonomists with the convenience of focusing on specific section, species-complexes and species. In the current study, a new species, Diaporthe camphorae, was introduced into the Diaporthe virgiliae species complex, based on morphology coupled to multilocus (ITS, cal, his3, tef1 and tub2) molecular analyses. The Diaporthe virgiliae species complex contains 5 species, viz. Diaporthe grandiflori, Diaporthe heterophyllae, Diaporthe penetriteum, Diaporthe virgiliae and Diaporthe zaofenghuang previously. As the number of Diaporthe specimens collected increases, the number and range of section, species-complexes and species may also increase and expand.

Comments 2: “fromtea” change to from tea

Response 2: We have changed it.

Comments 3: “Quercus rubr” change to Quercus rubra

Response 3: We have revised it.

Comments 4: The phyllogenetic tree is shown in the methodology, not in the results. The authors must put on the correct called of the text

Response 4: We have made some adjustments to make it easier for readers to read.

Comments 5: In Section: Description, “solitaryor” change to “solitary or”, “Orsubcylindrical” change to or subcylindrical, “toonly” separate these words, Cellshyaline separate these words. “Morphologically, the conidiophores of Paratubakia schimae are large than Paratubakia subglobosa” change to “ are largeR than”

Response 5: We have changed them.

We tried our best to improve the manuscript and made some changes marked in blue in revised paper which will not influence the content and framework of the paper. We appreciate for Editors/Reviewers’ warm work earnestly and hope the revision will meet with your approval. Once again, thank you very much for your comments and suggestions.

Kind regards,

Junzhi Qiu

E-mail address: junzhiqiu@126.com

Round 2

Reviewer 2 Report

See corrections and comments in the attached file.

See corrections and comments in the attached file.

Author Response

Please see revisions in the attached file.
